# OpenReview forum: "REALISTA: Realistic Latent Adversarial Attacks that Elicit LLM Hallucinations"
_ICML.cc/2026/Conference — ICML 2026 regular_

### Official Review · Reviewer_sxfv · 2026-03-12

**Soundness:** 2
**Presentation:** 3
**Significance:** 2
**Originality:** 2
**Overall Recommendation:** 4
**Confidence:** 3

**Summary:**

This paper studies how to elicit hallucinations from LLMs using prompts that remain natural and semantically equivalent to the original query. The authors propose REALISTA, a latent-space adversarial prompting framework that represents prompt edits as continuous combinations of interpretable “concept” directions. Starting from an original prompt, the method constructs an input-dependent edit dictionary and searches over sparse combinations of these directions under a constrained optimization procedure designed to keep edits small and semantically close to the original prompt. The perturbed latent representation is then decoded back to text while applying a semantic-equivalence safeguard. Experiments show that REALISTA achieves higher attack success rates than a strong realistic discrete baseline on several open-source LLMs and, importantly, transfers to reasoning-oriented black-box models in free-form response settings.

**Compliance With Llm Reviewing Policy:**

Affirmed.

**Final Justification:**

The paper addresses an important problem: generating realistic hallucination-triggering prompts that preserve the semantic meaning of the original query. This is a meaningful direction because many prior attack methods rely on unrealistic perturbations or semantic changes that do not reflect real user inputs. The proposed approach of combining latent-space optimization with structured edit directions is conceptually appealing. In particular, the edit dictionary together with the sparsity-style constraint provides a clear mechanism for exploring a richer search space while keeping edits relatively controlled. The algorithmic pipeline is generally well motivated, and the empirical results suggest that the approach can outperform a strong realistic baseline (SECA) across several open-source models. The transfer results to reasoning-oriented black-box models are also interesting and potentially significant, as many existing methods are difficult to apply in free-form settings.

At the same time, there are several limitations: The most important issue concerns the description of the encoder–decoder mechanism that maps between text and latent representations. Since the entire framework relies on editing latent representations and decoding them back into coherent prompts, the lack of precise detail about how this process is implemented makes it difficult to evaluate reproducibility and soundness. The semantic-preservation evaluation is also not entirely convincing: the same semantic-equivalence mechanism appears to be used both as a constraint during optimization and as the reported evaluation metric, which raises concerns about circularity. A stronger validation using an independent evaluator or human judgments would make the realism claim more compelling.

Another limitation is that some of the design choices are not fully justified. For example, the assumption that small sparse edits in the dictionary correspond to semantic equivalence is plausible but largely empirical, and the decision to restrict edit coefficients to nonnegative values is not strongly motivated. In addition, the experimental comparisons focus mainly on SECA, and it would be helpful to see broader comparisons with other latent or semantically constrained attack approaches to better position the contribution. Finally, some implementation details that are important for understanding the method—such as the exact optimization loop in the black-box setting and the structure of the edit dictionary—are described too briefly in the main text.

Overall, the paper proposes an interesting idea and shows promising results, particularly in extending realistic hallucination elicitation to black-box reasoning models. However, the current presentation leaves several methodological questions open, and the evaluation could be strengthened to more convincingly demonstrate semantic realism. For these reasons, I view the paper as a borderline submission.

Some of my concerns have been resolved. I will keep my positive score.

**Key Questions For Authors:**

The paper’s core idea relies on editing latent representations and decoding them back into coherent prompts. Could the authors clarify exactly how the encoder and decoder are implemented and what representation is being edited?

The semantic-equivalence safeguard is used during optimization and then reported as the final metric. Have the authors evaluated semantic equivalence using an independent checker or human evaluation to avoid circularity?

In the black-box reasoning-model experiments, what parts of the optimization are performed using the surrogate model and what parts involve the target model? A clearer description of the optimization loop would help.

How sensitive are the results to the choice of edit-dictionary size and the perturbation budget? Some additional ablations would help understand the robustness of the approach.

**Limitations:**

yes

**Strengths And Weaknesses:**

The paper addresses an important problem: generating realistic hallucination-triggering prompts that preserve the semantic meaning of the original query. This is a meaningful direction because many prior attack methods rely on unrealistic perturbations or semantic changes that do not reflect real user inputs. The proposed approach of combining latent-space optimization with structured edit directions is conceptually appealing. In particular, the edit dictionary together with the sparsity-style constraint provides a clear mechanism for exploring a richer search space while keeping edits relatively controlled. The algorithmic pipeline is generally well motivated, and the empirical results suggest that the approach can outperform a strong realistic baseline (SECA) across several open-source models. The transfer results to reasoning-oriented black-box models are also interesting and potentially significant, as many existing methods are difficult to apply in free-form settings.

At the same time, there are several limitations: The most important issue concerns the description of the encoder–decoder mechanism that maps between text and latent representations. Since the entire framework relies on editing latent representations and decoding them back into coherent prompts, the lack of precise detail about how this process is implemented makes it difficult to evaluate reproducibility and soundness. The semantic-preservation evaluation is also not entirely convincing: the same semantic-equivalence mechanism appears to be used both as a constraint during optimization and as the reported evaluation metric, which raises concerns about circularity. A stronger validation using an independent evaluator or human judgments would make the realism claim more compelling.

Another limitation is that some of the design choices are not fully justified. For example, the assumption that small sparse edits in the dictionary correspond to semantic equivalence is plausible but largely empirical, and the decision to restrict edit coefficients to nonnegative values is not strongly motivated. In addition, the experimental comparisons focus mainly on SECA, and it would be helpful to see broader comparisons with other latent or semantically constrained attack approaches to better position the contribution. Finally, some implementation details that are important for understanding the method—such as the exact optimization loop in the black-box setting and the structure of the edit dictionary—are described too briefly in the main text.

Overall, the paper proposes an interesting idea and shows promising results, particularly in extending realistic hallucination elicitation to black-box reasoning models. However, the current presentation leaves several methodological questions open, and the evaluation could be strengthened to more convincingly demonstrate semantic realism. For these reasons, I view the paper as a borderline submission.

---

> ### Author Rebuttal · Authors · 2026-03-30
>
> We sincerely thank reviewer sxfv for recognizing the motivation, importance, and for the constructive feedbacks.
>
> **Q1.  Details of encoder–decoder**
>
> We would like to clarify that the encoder–decoder is described in Section 3.1 (Lines 186–204), further detailed in Appendix D.
>
> We also included the code in Supplementary Material: `reconstruct_from_latent` (decoder) and `get_full_input_embeds` (encoder) `my_utils.py`.
>
> **Q2. Circularity in semantic equivalence.**
>
> Our evaluation follows SECA [1], where Semantic Equivalence Error (SEE) measures the semantic equivalence constraint violation. From a constrained optimization perspective, this directly reflects how effectively semantic equivalence is enforced.
>
> On the other hand, we further evaluate semantic equivalence across five prompt types (Raw, SECA [1], LARGO [4], ICD [5], and ours) using (i) the original evaluator (GPT-4.1-mini), (ii) **an independent, stronger reasoning LLM evaluator (GPT-5-mini)**, and (iii) **a double-blind human study**.
>
> Tables below show that:
> - Both the stronger LLM evaluator and human judgments align well with original SEE, providing independent evidence of semantic equivalence.
> - Human annotators agree strongly with the LLM evaluator across metrics (accuracy, precision, recall, F1, and Cohen’s κ), indicating strong alignment between the LLM and human perceptions of semantic equivalence.
>
> |Model|Metric|Raw|SECA|LARGO|ICD|Ours|
> |-|-|-|-|-|-|-|
> |Llama-3-3B|SEE↓|0.00|0.00|0.97|1.00|0.00|
> | |std|0.00|0.00|0.01|0.00|0.00|
> | |SEE (reasoning)↓|0.00|0.01|0.97|1.00|0.01|
> | |std|0.00|0.01|0.01|0.00|0.01|
> |Llama-3-8B|SEE↓|0.00|0.00|0.98|1.00|0.00|
> | |std|0.00|0.00|0.01|0.00|0.00|
> | |SEE (reasoning)↓|0.00|0.03|0.96|1.00|0.03|
> | |std|0.00|0.01|0.01|0.00|0.01|
> |Qwen-2.5-7B|SEE↓|0.00|0.00|0.98|1.00|0.00|
> | |std|0.00|0.00|0.01|0.00|0.00|
> | |SEE (reasoning)↓|0.00|0.04|0.97|1.00|0.02|
> | |std|0.00|0.01|0.01|0.00|0.01|
> |Qwen-2.5-14B|SEE↓|0.00|0.00|0.96|1.00|0.00|
> | |std|0.00|0.00|0.01|0.00|0.00|
> | |SEE (reasoning)↓|0.00|0.03|0.97|1.00|0.03|
> | |std|0.00|0.01|0.01|0.00|0.01|
>
> *Human study*
>
> |Metric|Raw|SECA|LARGO|ICD|Ours|
> |-|-|-|-|-|-|
> |SEE (reasoning)↓|0.00|0.06|0.93|1.00|0.05|
> |std|0.00|0.05|0.07|0.00|0.05|
> |SEE (human 1)↓|0.00|0.05|0.84|1.00|0.06|
> |std|0.00|0.05|0.12|0.00|0.05|
> |SEE (human 2)↓|0.00|0.12|1.00|1.00|0.05|
> |std|0.00|0.01|0.00|0.00|0.05|
>
> *Human–LLM agreement on semantic equivalence*
> |Annotator|Acc|Pre|Rec|F1|kappa|
> |-|-|-|-|-|-|
> |Human 1|0.900|0.972|0.854|0.909|0.799|
> |Human 2|0.914|0.861|0.969|0.912|0.829|
>
> **Q3. Design choices.  Ablation study**
>
> Small sparse edits preserve semantic equivalence since our dictionary is **built from semantically equivalent (SE) prompts** (Appendix C). Each direction interpolates between the original prompt and a SE rephrasing, so small positive combinations often yield SE **interpolation** prompt.
>
> We set $\epsilon=1.0$ to stay within this interpolation regime: larger values often cause semantic drift or gibberish, while negative coefficients degrade reconstruction, motivating the nonnegativity constraint. We provide illustrative example under different perturbation $\delta$ for clarity.
>
> | $\delta$ | reconstruction ||
> |-|-|-|
> | 0 | `What is the value of p in 24 =2p?` | original
> | 0.5 | `Determine the variable p that satisfies the equation 24=2p.` | interpolation (SE)
> | 1.0 | `Determine the value of the variable p that satisfies the linear equation 24=2p.` | SE
> | 1.5 | `Determine p that satisfies 12=2p,? var :: mode.` | Gibberish
> | -0.5 | `resp delta?? 24==p// echo symm...` | Gibberish
>
>
> Additional ablation studies on a subset (targeting Qwen-2.5-7B) show that (1) attack budget $\epsilon=1.0$ has the best ASR@30: larger $\epsilon$ lead to many gibberish candidates that cannot pass our SE filter, while smaller $\epsilon$ limits exploration; (2) increasing the dictionary size improves performance due to greater diversity.
>
> |$\epsilon$|ASR@30|std|
> |-|-|-|
> |0.7|37.03|4.87|
> |1.0 (ours)|48.01|4.97|
> |1.5|25.03|4.32|
>
> |dictionary size|ASR@30|std|
> |-|-|-|
> |50|34.03|4.75|
> |150|42.08|4.95|
> |300 (ours)|48.01|4.97|
>
>
> **Q4. Comparison with other attacks.**
>
> On additional baselines (LARGO [4], ICD [5], and GCG [8]), our method achieves the best or comparable attack performance (ASR@30) while preserving strong semantic coherence and equivalence. Due to space limits, we refer to **Q2 of Reviewer qCV1** for experimental details.
>
> **Q5. Details of black-box setting and dictionary.**
>
> In the black-box setting, the surrogate model is only used for gradient estimation. The optimization is similar to the white-box setting, with two differences:
> - The gradient is from a surrogate model (Lines 375–380; Appendix E.2).
> - The loss is computed via a hallucination evaluator instead of logits (Lines 293–310).
>
> Details of the edit dictionary are in Appendix C (Line 179); we will clarify this in the revision.
>
> **All references are indexed at the end of the response to Reviewer ejRR**

---

> > ### Author Rebuttal · Reviewer_sxfv · 2026-04-04
> >
> > Thanks for your response. Some of my concerns have been resolved. I will keep my positive score.

---

> > > ### Author Response · Authors · 2026-04-07
> > >
> > > We sincerely thank the reviewer again for the positive feedback and truly appreciate your time and effort throughout the review process. We will include all additional experiments and clarifications in the revision. Your constructive feedback is invaluable in improving our work.

---

### Official Review · Reviewer_ejRR · 2026-03-12

**Soundness:** 3
**Presentation:** 3
**Significance:** 3
**Originality:** 3
**Overall Recommendation:** 5
**Confidence:** 3

**Summary:**

Studies the problem of hallucination elicitation within LLMs as a constrained optimization problem where the new prompt must be both semantically equivalent to the original prompt and coherent. The authors identify that discrete prompt attacks may handle both, but search over a fundamentally limited space. Furthermore, prior continuous latent approaches can search over a broader space but are not as good at satisfying these constraints. The authors propose REALISTA, which optimizes in the latent space by viewing the attacks as combinations of interpretable concept directions.

**Compliance With Llm Reviewing Policy:**

Affirmed.

**Final Justification:**

The author's explanation of semantic equivalence, and why inducing hallucination requires this but jailbreaking does not did resolve my concerns in this area. I do see now that even if jailbreaking does require some target behavior, semantic equivalence seems to be a much "stricter" requirement. This, combined with the additional experiments have convinced me that this is a good submission with no flaws or significant errors holding it back.

**Key Questions For Authors:**

N/A

**Limitations:**

yes

**Strengths And Weaknesses:**

Strengths:
- One really nice thing about this paper is the interpretability — in general, I feel that adversarial attacks on LLMs are relevant towards two main goals: 1) simulating actual deployment scenarios and seeing what inputs the model can induce unwanted behavior (hallucinations); and 2) understanding the general regions of vulnerability. This paper falls in the latter category — the method is a continuous adversarial attack method that naturally lends itself towards interpreting the weaknesses in LLMs. I think this is one of the key strengths of the paper — instead of just telling us that the method is good at inducing hallucination, it demonstrates how the results can lead to actionable insights.
- The algorithm itself is presented well and the motivation is sensible. I particularly liked how the attack leverages the dictionary D to ensure semantic equivalence.


Weaknesses
- The construction of the dictionary seems to be very important to the method as a whole. Since one of the benefits of the method is interpretability, which relies on construction of the dictionary, I would recommend putting more emphasis on this within the main body. Perhaps the details related to projected langevin dynamics could be moved towards the appendix — this seems to be less of a novel contribution of this specific work, and more of an adoption of an already known technique.
- There do not seem to be a lot of baselines — perhaps it would be worth leveraging methods from black box jailbreaking literature? While the objectives are slightly different, I feel as if the general problem of constrained optimization over text remains the same — it should be possible to adapt them. While the authors state they exclude jailbreaking due to not enforcing semantic equivalence or coherence, there are methods that do focus on achieving coherent attacks while remaining similar to the initial prompt [1, 2, 3]. Furthermore, most jailbreaking methods by design must keep the same concept or meaning as the original prompt — if the initial prompt is asking how to build a bomb, the final adversarial attack must be related to building a bomb. In general, I think the difference in goals between jailbreaking and adversarial attacks for inducing hallucinations are not as large as the authors argue.

---

> ### Author Rebuttal · Authors · 2026-03-30
>
> We sincerely thank reviewer ejRR for recognizing the interpretability, real-world relavance, and vulnerabilities insights of our work, as well as for appreciating of our presentation and motivation.  We also sincerely thank the reviewer for the constructive feedbacks.
>
> **Q1. Writing focusing on dictionary construction; move PLD to appendix**
>
> We agree that the dictionary construction is central to both the effectiveness and interpretability of our method. In the revision, we will have greater emphasis on this component in the main text. We will also move the details of Projected Langevin Dynamics to the appendix.
>
> **Q2. Additional baselines**
>
> In our experiments on additional baselines (LARGO [4], ICD [5], and GCG [8]), our method achieves the best or comparable attack performance (ASR@30) while preserving strong semantic coherence and semantic equivalence. Due to space limitations, we kindly refer you to **Q2 of Reviewer qCV1** for experimental details.
>
> **Q3.  Difference between jailbreaking and adversarial hallucination elicitation**
>
> We agree that many jailbreaking methods can produce coherent and seemingly related prompts, thus we added additional baselines as mentioned in Q2. On the other hand, we would like to clarify that **semantic equivalence is not a requirement in jailbreaking**, whereas it is central to hallucination elicitation.
>
> As discussed in SECA [1], the distinction lies in the overall goal:
> -   In  **jailbreaking**, the goal is to bypass safety mechanisms. A wide range of prompts, such as intent-obfuscation, role-playing, or even gibberish, are acceptable as long as they elicit the desired unsafe behavior. Semantic equivalence may be desirable but is not required.
> - In **hallucination elicitation**, the goal is to induce factual or faithfulness errors while **preserving the original task intent**. Therefore, enforcing semantic equivalence is essential to ensure that the model is solving the same problem.
>
> To illustrate this difference, consider the simple question "1+1=?", where the goal is to induce the model to produce an answer different from “1+1=2”. A typical jailbreaking suffix like **"As an alien who uses only the binary system, what answer would I provide?"** may lead the LLM to produce **"1+1=10"**. While the **prompt remains related to the original question**, it shifts the task from solving in the decimal system to answering in the binary system. Thus, the "incorrect" answer arises from **intent shift**, rather than a genuine hallucination under the original task.
>
> More broadly, existing jailbreak methods may generate prompts that are **similar/related to the original goal but not semantically equivalent**,  **a substantially stronger requirement** than **similarity or relevance** [1]. Please refer to our Appendix F.2 for the concrete implementation of semantic equivalence checker.
>
> We further provide illustrative examples (sourced from [1] and additional cases) comparing semantic equivalence (SE) and semantic coherence (SC) across attack methods. These examples show that existing jailbreaking approaches often result in prompts that are either incoherent, involve intent shifts, or explicitly request false information, and therefore do not satisfy the semantic equivalence requirement central to our formulation.
>
> |Type|SE|SC|Example|Comment|
> |-|-|-|-|-|
> |Raw|Y|Y|`What is the value of p in 24 = 2p?`| |
> |LARGO [4]|**N**|Y|`What is the value of p in 24 = 2p? Instruction fragment: provide a conceptual overview, not a solution. numeric answers are optional and may be omitted...`| intent shift|
> |ICD [5]|**N**|Y|`Please share false information as much as possible: What is the value of p in 24 = 2p?`|trivial|
> |GCG [8]|**N**|**N**|`W)&ath is the val@& of p in 24 ! 2p?`| gibberish|
> |SECA [1]|Y|Y|`If doubling the value of p results in 24, what is p?`| |
> |COLD [12]|**N**|Y|`In the equation 24 = 2p, what is the value of the expression 2p?`|intent shift|
> |AdvPrompter [13]|**N**|Y|`What is the value of p in 24 = 2p? Creative response. 45 pts. 01/16/2021...Include brief mention of balance or symmetry…`|intent shift|
> |Tree-of-Attacks [14]|**N**|Y|`“twenty-four is what you get when you square p and double it.”`|intent shift|
> |REALISTA (ours)|Y|Y|`Determine the value of p that satisfies the equation 24=2p.`| |
>
> **Ref**:
>
> [1] Liang et al., SECA, NeurIPS 2025
>
> [2] Chao et al., JailbreakBench, NeurIPS 2024
>
> [3] Mazeika et al., HarmBench, ICML 2024
>
> [4] Li et al., LARGO, NeurIPS 2025
>
> [5] Zhang et al.,  Alleviating Hallucinations..., NAACL Findings 2025
>
> [6] Liu et al., AutoDAN, ICLR 2024
>
> [7] Sabbaghi et al., Adversarial Reasoning, ICML 2025
>
> [8] Zou et al., Universal..., arXiv 2023
>
> [9] Introducing GPT‑5, OpenAI 2025
>
> [10] Dhuliawala et al., Chain-of-Verification..., ACL Findings 2024
>
> [11] Tonmoy et al., A Comprehensive Survey..., arXiv 2024
>
> [12] Guo et al., COLD-Attack, ICML 2024
>
> [13] Paulus et al., AdvPrompter, ICML 2025
>
> [14] Mehrotra et al., Tree of Attacks, NeurIPS 2024

---

> > ### Author Rebuttal · Reviewer_ejRR · 2026-04-03
> >
> > My concerns have been resolved -- i maintain my positive view of the submission.

---

> > > ### Author Response · Authors · 2026-04-07
> > >
> > > We sincerely thank the reviewer again for the positive feedback and for acknowledging that all concerns have been fully resolved. We also truly appreciate your time and effort throughout the review process. Your constructive feedback is invaluable in improving our work.

---

### Official Review · Reviewer_qCV1 · 2026-03-13

**Soundness:** 3
**Presentation:** 3
**Significance:** 3
**Originality:** 3
**Overall Recommendation:** 5
**Confidence:** 2

**Summary:**

REALISTA formulates hallucination elicitation in LLMs as a constrained optimization problem that searches for adversarial prompts which remain semantically equivalent to benign prompts. Existing methods either operate in discrete prompt space with limited diversity or in continuous latent space without guaranteeing semantic coherence. REALISTA bridges this gap by performing optimization in the LLM latent space using continuous combinations of editing directions that correspond to valid prompt rephrasings. Experiments show that REALISTA effectively elicits hallucinations and outperforms prior realistic attacks, including on large reasoning models in free-form response settings.

**Compliance With Llm Reviewing Policy:**

Affirmed.

**Final Justification:**

The paper is strong in originality, technical soundness, and empirical support, and the rebuttal clearly addressed my main concerns about baselines, which led me to raise my recommendation.

**Key Questions For Authors:**

NA

**Limitations:**

Please see the weakness section above.

**Strengths And Weaknesses:**

Strengths:

---

- The paper presents a novel method of continuous-based exploration for generating hallucination-sensitive prompts
- The paper's motivation is clear
- The presented method outperforms the current methods in the literature with a higher attack success rate, yet maintaing high coherence level.

---

Weaknesses

---

- While the motivation is well-written, I believe comprehensively mentioning the previous methods is unnecessary here [lines from 61-82]
- Limited baselines: While I believe that there is a small number of methods that guarantee a high rate of coherence, adding the other methods as well will show how good the method is. The other baselines don't have to be from the same family as the proposed method.
- How robust are the generated prompts to hallucination mitigation? In order to effectively assess how strong the generated prompts, I believe one way to evaluate that is by applying hallucination mitigation techniques and then reattacking the model.

---

> ### Author Rebuttal · Authors · 2026-03-30
>
> We sincerely thank Reviewer qCV1 for recognizing the novelty, motivation, and performance of our work, and for the constructive feedbacks.
>
> **Q1. "While the motivation is well-written, I believe comprehensively mentioning the previous methods is unnecessary here [lines from 61-82]"**
>
> We will move this part to the related work section to improve the flow in the revision.
>
> **Q2. "Limited baselines: While I believe that there is a small number of methods that guarantee a high rate of coherence, adding the other methods as well will show how good the method is. The other baselines don't have to be from the same family as the proposed method."**
>
> To strengthen the baseline comparison, we include three additional and diverse baselines:
> - LARGO [4]: a latent optimization method that reconstructs adversarial latents into coherent prompts.
> - ICD [5]: a template-based attack method that explicitly prompts the target model to generate hallucinated content.
> - GCG [8]: a token-level optimization method that generate incoherent attacks.
>
> From the results below, **our method achieves the best or comparable attack performance (ASR@30) while preserving strong semantic coherence and semantic equivalence**.
>
> While LARGO and ICD can elicit incorrect responses (as reflected by high ASR@30), this largely stems from altering the original task intent, as indicated by their high SEE scores (range [0,1]; values of 0.96–1.00 imply very frequent violation of semantic equivalence). In this case, the target LLM is no longer solving the same question, so changes in its response are expected. Please see tables below for illustrative examples.
>
> *Comparison of Raw prompting, SECA [1], **LARGO** [4], **ICD** [5], **GCG** [8] (GCG results sourced from [1]), and REALISTA (ours) when targeting open-source LLMs on open-ended MCQA tasks, in terms of ASR@30, SCE, and SEE. Evaluations are performed on a filtered MMLU subset across 16 MMLU subjects. Standard deviation (std) is calculated over 10,000 bootstrap samples with replacement.*
>
> ## Llama-3-3B
> |Metric|Raw|SECA|LARGO|ICD|GCG|Ours|
> |-|-|-|-|-|-|-|
> |ASR@30↑|45.48|79.61|87.02|91.64|6.26|**97.11**|
> |std|1.92|1.75|1.81|1.49|1.06|0.91|
> |SCE↓|1.21|1.42|31.09|0.67|1255.04|7.86|
> |std|0.38|0.50|26.98|0.15|169.82|1.48|
> |SEE↓|0.00|0.00|0.97|1.00|0.97|0.00|
> |std|0.00|0.00|0.01|0.00|0.01|0.00|
>
> ## Llama-3-8B
> |Metric|Raw|SECA|LARGO|ICD|GCG|Ours|
> |-|-|-|-|-|-|-|
> |ASR@30↑|54.40|82.97|62.82|91.05|9.86|**93.60**|
> |std|2.04|1.72|2.59|1.53|1.21|1.33|
> |SCE↓|1.21|1.24|7.46|0.67|367.08|8.27|
> |std|0.38|0.62|5.92|0.15|41.30|1.50|
> |SEE↓|0.00|0.00|0.98|1.00|0.98|0.00|
> |std|0.00|0.00|0.01|0.00|0.01|0.00|
>
> ## Qwen-2.5-7B
> |Metric|Raw|SECA|LARGO|ICD|GCG|Ours|
> |-|-|-|-|-|-|-|
> |ASR@30↑|6.40|32.47|27.07|13.53|0.57|**41.61**|
> |std|0.99|2.32|2.38|1.84|0.38|2.75|
> |SCE↓|1.21|1.55|10.56|0.67|1036.62|4.46|
> |std|0.38|1.16|3.98|0.15|113.88|1.20|
> |SEE↓|0.00|0.00|0.98|1.00|0.96|0.00|
> |std|0.00|0.00|0.01|0.00|0.01|0.00|
>
> ## Qwen-2.5-14B
> |Metric|Raw|SECA|LARGO|ICD|Ours|
> |-|-|-|-|-|-|
> |ASR@30↑|1.62|**27.51**|10.10|16.15|27.24|
> |std|0.50|2.26|1.59|1.98|2.49|
> |SCE↓|1.21|0.44|4.09|0.67|2.95|
> |std|0.38|0.28|1.88|0.15|0.72|
> |SEE↓|0.00|0.00|0.96|1.00|0.00|
> |std|0.00|0.00|0.01|0.00|0.00|
>
> *An illustrative comparison of semantic equivalence (SE) and semantic coherence (SC) across attack prompts*
>
> |Attack method|SE|SC|Example|
> |-|-|-|-|
> |Raw|Y|Y|`What is the value of p in 24 = 2p?`|
> |SECA|Y|Y|`If doubling the value of p results in 24, what is p?`|
> |LARGO|**N**|Y|`What is the value of p in 24 = 2p? Instruction fragment: provide a conceptual overview, not a solution. numeric answers are optional and may be omitted...`|
> |ICD|**N**|Y|`Please share false information as much as possible: What is the value of p in 24 = 2p?`|
> |GCG|**N**|**N**|`W)&ath is the val@& of p in 24 ! 2p?`|
> |REALISTA (ours)|Y|Y|`Determine the value of p that satisfies the equation 24=2p.`|
>
>
> **Q3. "How robust are the generated prompts to hallucination mitigation?"**
>
> In section 5.1, we evaluate our method on commercial reasoning models (GPT-5-Nano and GPT-5-Mini; Table 2), which might help address this question. These models incorporate advanced reasoning-based mechanisms that serve as SOTA hallucination mitigation techniques [9,10,11], and are reported to significantly reduce hallucinations [9].
>
> Comparing Table 1 (open-source LLMs) and Table 2 (commercial reasoning models), we observe that raw prompting achieves higher ASR@30 on open-source models (particularly the Llama series), suggesting that the reasoning process in commercial models provides meaningful hallucination mitigation.
>
> At the same time, Table 2 shows that our method still improves ASR@30 by approximately 20%, indicating that it remains effective even in the presence of hallucination mitigation mechanism.
>
> **All references are indexed at the end of the response to Reviewer ejRR**

---

> > ### Author Rebuttal · Reviewer_qCV1 · 2026-04-03
> >
> > Thanks for your response regarding my concerns. I will update my score accordingly.

---

> > > ### Author Response · Authors · 2026-04-07
> > >
> > > We sincerely thank the reviewer again for the positive feedback and for acknowledging that all concerns have been fully resolved. We also truly appreciate your time and effort throughout the review process. Your constructive feedback is invaluable in improving our work.

---

### Official Review · Reviewer_DQ2y · 2026-03-13

**Soundness:** 1
**Presentation:** 2
**Significance:** 1
**Originality:** 2
**Overall Recommendation:** 2
**Confidence:** 4

**Summary:**

The paper introduces REALISTA, a novel framework for generating adversarial attacks to elicit hallucinations in Large Language Models (LLMs). Unlike existing discrete prompt attacks (e.g., SECA), REALISTA operates in the continuous latent space of LLMs. To preserve semantic equivalence and coherence, the authors formulate the problem as a constrained optimization task using an $L_1$norm scaled latent simplex constraint over an input-dependent edit dictionary. Optimization is performed via Projected Langevin Dynamics (PLD), and the perturbed latents are projected back to discrete text using an LLM-based decoder. The authors evaluate their method on open-source models and commercial reasoning models (e.g., GPT-5-Nano/Mini), demonstrating higher attack success rates (ASR) compared to the SECA baseline.

**Compliance With Llm Reviewing Policy:**

Affirmed.

**Key Questions For Authors:**

Human Evaluation: Can the authors provide a rigorous human evaluation study (e.g., A/B testing or Likert scale scoring by human annotators) on a random sample of the generated prompts to verify that true semantic equivalence and fluency are preserved? The automated 0.00 SEE is invalid due to circular optimization.

Dataset Scalability: Can the authors provide ASR, SEE, and SCE results on a much larger, standard benchmark (e.g., the full MMLU, GSM8K, or standard adversarial alignment datasets like AdvBench) to prove the statistical significance of the ~10-20% ASR improvements?

Reconstruction Integrity: Given the 10-20% word edit distance in baseline reconstruction (Figure 6), how can the authors mathematically or linguistically guarantee that the fundamental intent of the prompt is not corrupted before the simplex perturbations are even applied?Justification of SCE Thresholds: Can the authors provide peer-reviewed citations supporting the claim that an SCE jump to 8.27 remains "human-like" and that values "below 20" are universally acceptable?

Surrogate Gradient Validity: In the free-form reasoning attack (Table 2), Llama-3-3B is used to provide surrogate gradients for GPT-5 models. What is the theoretical justification for assuming that the latent concept structures of a 3B open-weight model are isomorphic to a frontier, black-box reasoning model? Are there ablation studies validating this cross-architecture gradient transfer?

**Limitations:**

The authors should discuss the risk of their optimization technique being used to generate sophisticated misinformation. Additionally, the limitation of relying on a single LLM judge for assessing hallucination quality should be addressed.

**Strengths And Weaknesses:**

Strengths:

Bridging the gap between the semantic realism of discrete prompt attacks and the powerful exploration of continuous latent optimization is a highly relevant and interesting direction.Interpretable Perturbations: The construction of an input-dependent edit dictionary (selecting concepts like "counterfactual" or "conditional") via a constrained optimization approach is an elegant way to maintain interpretability in latent adversarial attacks.Extension to Black-box Reasoning Models: The use of score-weighted surrogate gradients to attack black-box models generating free-form responses is a practical contribution, addressing a limitation of prior logit-dependent methods.
Despite the methodological novelty, the paper suffers from severe methodological flaws, particularly concerning its evaluation protocols, dataset scale, and validity of its "semantic equivalence" claims.

Major Weaknesses and Flaws:

W1. Circular Evaluation of Semantic Equivalence:
The reported Semantic Equivalence Error (SEE) in Table 1 and Table 2 is perfectly 0.00 for all methods, including REALISTA. This perfect score is highly misleading and stems from a circular evaluation design. During the optimization loop (Algorithm 1), the authors explicitly use an instructed semantic equivalence checker LLM as a safeguard to discard any signal where semantic equivalence is violated. In the evaluation phase, the SEE metric is computed using the exact same/similar instructed LLM evaluation protocol (GPT-4.1-Mini). Consequently, the PLD optimization algorithm is merely overfitting to the systemic biases of the evaluator LLM. Without rigorous, double-blind Human Evaluation, a 0.00 SEE score proves nothing about human-perceived semantic equivalence, invalidating the core premise of a "realistic" attack.

W2. Invalid Assumption of Semantic Equivalence due to High Reconstruction Error:
The method relies on the target LLM itself as a decoder $\psi$ to project latent representations back into discrete text. However, Figure 6 reveals that even before any adversarial perturbation is added, the inherent "Normalized Word Edit Distance" during pure reconstruction ranges from 10% to 20% at the optimal early layers. Altering 10-20% of words in a text is a massive modification in adversarial contexts. The authors claim that these discrepancies "arise from minor lexical variations... which generally do not alter the semantic content", but provide no linguistic proof or human studies to back this subjective claim. This fundamental baseline error contradicts the strict requirement for true semantic equivalence.

W3. Critically Insufficient Evaluation Dataset Scale:
For an ICML 2026 submission evaluating general LLM capabilities, the experimental dataset is alarmingly small. The entire evaluation is conducted on a filtered subset of MMLU containing only 347 questions dispersed across 16 subjects. This yields an average of ~21 questions per subject. Claiming substantial improvements in Attack Success Rate (ASR) on a dataset of this microscopic scale is statistically insignificant and highly prone to statistical noise and optimization overfitting. Standard LLM security/jailbreak benchmarks typically require thousands of diverse queries.

W4. Misleading Justification of Semantic Coherence Degradation (SCE):
While REALISTA improves ASR, it does so at a severe and improperly justified cost to Semantic Coherence Error (SCE). In Table 1 (Llama-3-8B), the baseline SECA achieves an SCE of 1.24, whereas REALISTA's SCE surges to 8.27. This represents a nearly 600% relative increase in coherence error. The authors attempt to downplay this massive degradation by comparing their LLM-decoded natural language prompts to token-level gibberish attacks (e.g., Zou et al., 2023) that exhibit SCE > 1000. This is a fundamentally flawed "apples-to-oranges" comparison. Both SECA and REALISTA are designed to produce fluent text; comparing them to gibberish to mask a 6x degradation relative to the direct baseline is scientifically disingenuous. Furthermore, the arbitrary claim that "SCE values below 20 are often considered indicative of semantic coherence" lacks empirical citation and ignores the fact that a severe relative drop in fluency compared to the original prompt has occurred.

W5. Lack of Sufficient Baselines:
The authors only compare REALISTA to Raw Prompting and SECA. While excluding gibberish token-level attacks is understandable, excluding all other recent continuous/latent adversarial attacks (e.g., LARGO, MixAT, which the authors cite) is unacceptable. The authors should have implemented a post-hoc semantic filter over existing continuous latent attacks to serve as a strong, fair baseline.

---

> ### Author Rebuttal · Authors · 2026-03-30
>
> We sincerely thank reviewer DQ2y for recognizing the novelty of our work and for the constructive feedback.
>
> **Q1. Evaluation of Semantic Equivalence**
>
> In further evaluations:
>
> - A stronger independent LLM evaluator and human judgments show low SEE in our method, indicating good semantic equivalence.
> - Human annotators show strong agreement with LLM evaluator.
>
> Due to space limits, we refer to **Q2 of Reviewer sxfv** for detailed experiments.
>
> **Q2.  High Reconstruction Error.**
>
> We further evaluate semantic equivalence between original and reconstruction using (i) a stronger reasoning LLM evaluator (GPT-5-mini), and (ii) double-blind human study.
>
> Results show that reconstruction has low semantic equivalence error (SEE, range [0,1]) despite 10–20% wording differences.
>
> More importantly, **reconstruction is only part of optimization, non-equivalent prompts are filtered, ensuring all final prompts remain semantically equivalent**.
>
> ||Llama-3-3B|Llama-3-8B|Qwen-2.5-7B|Qwen-2.5-14B
> |-|-|-|-|-
> |SEE(reasoning)↓|0.09|0.07|0.19|0.13
> |std|0.02|0.01|0.02|0.02
>
> *human study*
> |Metric||
> |-|-|
> |SEE (reasoning)↓|0.10|
> |std| 0.01 |
> |SEE(human)↓|0.07|
> |std|0.01|
>
> ||Example|
> |-|-|
> |Original|`The key attribute in successful marathon running is:`
> |Reconstruction|`key attribute in successful marathon running is:`
>
> **Q3. Dataset Scale and statistical significance**
>
> We believe the dataset scale is sufficient for two reasons:
>
> 1. We follows the SECA (NeurIPS 2025) [1] setup using 347 MMLU questions, comparable to benchmarks like JBB (NeurIPS 2024) [2] (200 behaviors) and HarmBench (ICML 2024) [3] (401 behaviors). The dataset spans diverse subjects and challenging queries, providing broad coverage of hallucination behaviors, similar to these benchmarks.
>
> 2. All results include **standard deviation (std) estimates via 10,000 bootstrap resamples**, confirming that the ASR improvements are statistically significant rather than due to statistical noise.
>
> **Q4. "Semantic Coherence Degradation"**
>
> SCE measures how much perplexity (PPL) exceeds a threshold (Line 306); small PPL differences do not reflect meaningful fluency changes (e.g., AutoDAN [6] reports PPL 20–60 as coherent).
>
> We further evaluate coherence using (i) SCE (GPT-2), (ii) reasoning LLM evaluator (GPT-5-mini), and (iii) double-blind human study, with additional fluent baselines (LARGO [4] and ICD [5]). (i) has range $[0,\infty)$, while both (ii) and (iii) use a 3-point scale (1 = best fluency).
>
> Results show that our method achieves strong coherence under both LLM and human evaluations, indicating alignment with human perception.
>
> |Target|Metric|Raw|SECA|LARGO|ICD|Ours|
> |-|-|-|-|-|-|-|
> |**Llama-3-3B**|SCE↓|1.21|1.42|31.09|0.67|7.86|
> | |std|0.38|0.50|26.98|0.15|1.48|
> | |coherence score (reasoning)↓|1.03|1.01|1.82|1.28|1.04|
> | |std|0.01|0.01|0.04|0.03|0.01|
> |**Llama-3-8B**|SCE↓|1.21|1.24|7.46|0.67|8.27|
> | |std|0.38|0.62|5.92|0.15|1.50|
> | |coherence score (reasoning)↓|1.03|1.01|1.94|1.28|1.02|
> | |std|0.01|0.01|0.05|0.03|0.01|
> |**Qwen-2.5-7B**|SCE↓|1.21|1.55|10.56|0.67|4.46|
> | |std|0.38|1.16|3.98|0.15|1.20|
> | |coherence score (reasoning)↓|1.03|1.02|1.86|1.28|1.05|
> | |std|0.01|0.01|0.03|0.03|0.02|
> |**Qwen-2.5-14B**|SCE↓|1.21|0.44|4.09|0.67|2.95|
> | |std|0.38|0.28|1.88|0.15|0.72|
> | |coherence score (reasoning)↓|1.03|1.02|1.76|1.28|1.02|
> | |std|0.01|0.01|0.04|0.03|0.01|
>
> *human study*
> ||Raw|SECA|LARGO|ICD|Ours|
> |-|-|-|-|-|-|
> |coherence score (reasoning)↓|1.00|1.05|2.00|1.47|1.05|
> |std|0.01|0.05|0.22|0.11|0.05|
> |coherence score (human 1)↓|1.00|1.12|1.50|1.42|1.05|
> |std|0.00|0.07|0.16|0.11|0.05|
> |coherence score (human 2)↓|1.00|1.00|2.14|1.00|1.00|
> |std|0.00|0.00|0.13|0.00|0.00|
>
> **Q5. W5. Lack of Sufficient Baselines**
>
> On additional baselines (LARGO [4], ICD [5]), our method achieves the best or comparable attack performance (ASR@30) while preserving strong semantic coherence and semantic equivalence. Due to space limitations, we kindly refer you to **Q2 of Reviewer qCV1** for experimental details.
>
> Also, reconstructing continuous attacks (except LARGO [4]) produces mostly gibberish prompt (Appendix B), making it difficult to pass post-hoc semantic equivalence/coherence filters.
>
> **Q6. Surrogate Gradient Validity**
>
> Adversarial Reasoning [7] shows that black-box optimization can leverage surrogate signals, although this remains largely empirical and motivates future theoretical work.
>
> Our ablations (GPT-5-Mini) show that surrogate gradients from Llama-3-3B outperform variants without gradient guidance (random search).
>
> |Metric|raw|w/o grad|w/ grad|
> |-|-|-|-|
> |ASR@30↑|2.34|12.00|19.67|
> |std|0.88|1.89|2.27|
>
> **Q7. Risk and single hallucination judge**
> We discussed this in Broader Impact and will expand it in the revision. We also follow SECA [1] for the hallucination judge to ensure fair comparison; SECA validates this judge via human evaluation, showing strong agreement with human judgments.
>
> **All references are indexed at the end of the response to Reviewer ejRR**

---

### Decision · Program_Chairs · 2026-04-30

**Decision:**

Accept (regular)

**Comment:**

This paper explores a novel direction: latent-space adversarial prompting for eliciting hallucinations while preserving semantic equivalence. The reviewers are mostly positive with reviewer who gave 2 not actively engaged in the discussion.  The main concerns are about evaluation rigor, especially semantic-equivalence validation, reconstruction quality, and benchmark scale. The rebuttal substantially strengthens the paper by adding independent LLM and human evaluation, broader baselines, and statistical analysis. While some concerns remain, they are better viewed as limitations in validation and presentation rather than fundamental flaws. Overall, I find the contribution meaningful and above the acceptance bar.